# Geoadditive Quantile Regression Model for Sewer Pipes Deterioration Using Boosting Optimization Algorithm

**Ngandu Balekelayi and Solomon Tesfamariam** *

School of Engineering, University of British Columbia, Kelowna, BC V1V 1V7, Canada; ngbalek@mail.ubc.ca
* Correspondence: Solomon.Tesfamariam@ubc.ca

**Abstract:** Proactive management of wastewater pipes requires the development of deterioration models that support maintenance and inspection prioritization. The complexity and the lack of understanding of the deterioration process make this task difficult. A semiparametric Bayesian geoadditive quantile regression approach is applied to estimate the deterioration of wastewater pipe from a set of covariates that are allowed to affect linearly and nonlinearly the response variable. Categorical covariates only affect linearly the response variable. In addition, geospatial information embedding the unknown and unobserved influential covariates is introduced as a surrogate covariate that capture global autocorrelations and local heterogeneities. Boosting optimization algorithm is formulated for variable selection and parameter estimation in the model. Three geoadditive quantile regression models (5%, 50% and 95%) are developed to evaluate the band of uncertainty in the prediction of the pipes scores. The proposed model is applied to the wastewater system of the city of Calgary. The results show that an optimal selection of covariates coupled with appropriate representation of the dependence between the covariates and the response increases the accuracy in the estimation of the uncertainty band of the response variable. The proposed modeling approach is useful for the prioritization of inspections and provides knowledge for future installations. In addition, decision makers will be informed of the probability of occurrence of extreme deterioration events when the identified causal factors, in the 5% and 95% quantiles, are observed on the field.

**Keywords:** Bayesian; geoadditive; quantile; P-splines; boosting optimization

## 1. Introduction

Wastewater systems collect and transport household, commercial and industrial used water through a complex underground sewer pipe network to treatment plant facilities [1,2]. The need to maintain the required level of service and protect the environment led municipalities to actively monitor and improve their systems based on actual and future performance indicators. Budget restrictions and large size of modern wastewater systems, however, limit the ability of municipalities to inspect and assess the condition of all pipes in their networks [3–6]. For example, in Canada, only 23% of large collection pipes have complete inspection data and the condition of the remaining proportion is estimated based on proxy information, such as age, soil types and expert opinion [7]. Existing deterioration models focus on relating the mean of the response variable to a given (linear, nonlinear) combination of the covariate effects [8,9]. The use of such models implies that the estimated effects describe the average condition of pipes. However, it is of interest to analyze the quantiles of the response, such as 5% and 95% quantiles, which provide

information about the occurrence of extreme deterioration events (e.g., new pipes in worst structural condition or old pipes in excellent condition). Structured additive quantile regression coupled with the boosting algorithm (i.e., an optimization algorithm that aims at minimizing an expected loss criterion by stepwise updating of an estimator according to steepest gradient descent of the loss criterion) will simultaneously address the selection of relevant factors in the model and estimate model parameters in a single procedure [10].

Various mathematical models (physical, statistical, machine learning) have been proposed to model sewer pipes deterioration [11]. Lack of understanding of the complex physical process of pipe failure coupled with the limited data availabilities hinders use of physical models [12,13]. Artificial intelligence constitute an alternative approach that allow nonlinear modeling and fits well while the inherent bias and sparseness of sewer inspection data may prohibit the application of conventional statistical models [14]. The necessity of specification of internal model architecture, the need of large training sets and the sensitivity to imbalanced datasets limits their usage [15]. Furthermore, the causal relationship is not explicitly expressed. Statistical models are the most used techniques [16,17]. Regression models, a subcategory of statistical models, allow the identification of the causal relationship between the individual sewer attributes and its internal condition state [18]. Different regression approaches have been developed including linear regression [19,20] and nonlinear regression [9,21]. Despite efforts made by municipalities to collect information and build extensive and accurate databases of their wastewater systems, only physical characteristics of sewer pipes (e.g., length, diameter), are the most used attributes in statistical deterioration models for wastewater pipes [17,22–27]. However, various environmental and operational variables impact the structural safety of pipes [28–30]. Accuracy and reliability of existing regression based sewer deterioration models can be enhanced if the following shortfalls are addressed: (1) an oversimplification of the physical process (e.g., linear relationship between the covariates and the condition of pipes), (2) not all the required covariates are available nor included in deterioration models [30–33], (3) the spatial location of pipes that can inform about unknown and unobserved covariates is not considered [6,34,35].

Recent development of Gaussian structured additive regression models allows not only the inclusion of nonlinear effects of the covariates on the response variables but also the consideration of possible spatial autocorrelation to enhance predictive capabilities of wastewater deterioration models [9,13]. The geospatial location of pipes is introduced in the model to capture the information about unobserved covariates in the form of single surrogate covariate representing the district where the pipe is buried. A Gaussian Markov Random field (GMRF) approach allows pipes in neighborhood districts to have similar deterioration pattern, while a random effect parameter is applied to represent heterogeneities in pipes deterioration [34].

Davies et al. [30] listed 33 potential factors that might affect the deterioration of sewer pipes. However, all these factors are not always available in municipality databases to support the analysis of deterioration of sewer pipes. A parsimonious and interpretable model consisting of relevant factors represented at appropriate complexity is required. Statistically significant variables, in regression models, are identified based on heuristic techniques such as stepwise selection that are not efficient. In addition, in frequentist-based statistical models, the heuristic approach has a risk of overconfident estimation of the output variable [36,37]. Kabir et al. [38] applied Bayesian Model Averaging to identify significant covariates in a sewer system before applying a Bayesian logistic regression to predict the structural condition of pipes. Mohammadi et al. [39] considered thirteen variables including environmental factors to classify sanitary sewers. They proposed a gradient decision boosting algorithm to classify the pipes into two groups (good and critical pipes). The binary output of their these models limits its their use in the decision making. Laakso et al. [31] applied the Boruta algorithm to assess the importance of variable in a binary random forest sewer pipe deterioration model. Quantile regression aims at determining the influence of regressors on the conditional quantiles of the dependent variable which do not have necessarily

a Gaussian distribution [40]. The advantage of the analysis of quantiles in sewer pipe deterioration is the assessment of the tails of the distribution of the response. For instance, the 5% quantile contains old pipes in good condition and the 95% quantile contains new pipes that have the high likelihood of failure. Both extreme cases influence the expected mean models. Furthermore, the pipes attributes that are responsible of the appearance of outliers may differ is each quantile. The identification of these factors through appropriate modeling of the deterioration process will allow decision makers to capture extreme deterioration patterns and reduce the uncertainty in the pipes useful life calculations. Boosting algorithm has been successfully applied for variable selection and model choice to estimate the time to failure of water pipes [41] and the classification of sanitary sewers [39]. Application of quantile regression coupled to boosting algorithm to simultaneously minimize an empirical risk defined as a loss function while estimating model parameters and iteratively select informative regressors are found in [10,42,43]. Balekelayi and Tesfamariam [42] applied quantile regression to capture the frequency of water pipes' failure in the city of Calgary. The boosting optimization algorithm has the advantage over other machine learning, such as random forests, artificial neural networks, that result in black-box predictions and do not allow quantification of partial influence of covariates on the response variable [37].

In this paper, a Structured Quantile Additive Regression (SQRM) using a boosting optimization algorithm for the selection of covariates is proposed. The approach presented in [9,42] is extended with the integration of linear components for continuous covariates, i.e., the continuous covariates are allowed to have two components including the linear and the nonlinear effects modeled following the ordinary least square and the penalized splines nonlinear smooth functions respectively. Only linear effect is considered for categorical variables. A Bayesian inference based on Markov Chain Monte Carlo simulation is used to capture uncertainty propagation in the definition of base learners' models. The response data, defined as a continuous variable (pipe scores obtained after grading observed defects following a given protocol as described in [44]), is iteratively estimated through an optimized process of learners selection. The Bayesian-based evaluation of the effect of individual factors coupled with the optimization-based selection of the most informative factors increases the accuracy in the quantile prediction and allows the decision makers to confidently quantify the uncertainty in the deterioration model.

This paper is organized as follow: the next section gives the mathematical formulation of the proposed methodology. The description of the case study, the available data and the proposed model follow. Results and discussions are presented before a conclusion is presented in the last section.

## 2. Mathematical Formulation

The proposed framework is presented in Figure 1. After the selection of the wastewater system, the steps to compute SQRM are:

- First, collection of the available data including inspection data (e.g., observed defects, their severity, their orientation and their location), pipe characteristics (e.g., length, diameter, material type, buried depth and slope) and maintenance data (e.g., number of previous flushes, number of previous reported backups). If available, the operational data (e.g., dry weather flow, wet weather flow, inflow infiltration) are also collected.
- Second, data processing where variables are classified into categorical (e.g., material) and continuous (e.g., diameter). The response variable is the internal condition of the pipe that is expressed in terms of score obtained after rating the observed defects following a grading protocol (e.g., Water Resource Center: WRc). The model equation is developed, and the base learners' priors are selected.
- Third, the last step is simulation and model performance evaluation. The number of iterations and step-length are set to the same values in both quantiles. The validated models are used for predictions and the effect of the selected covariates are examined.

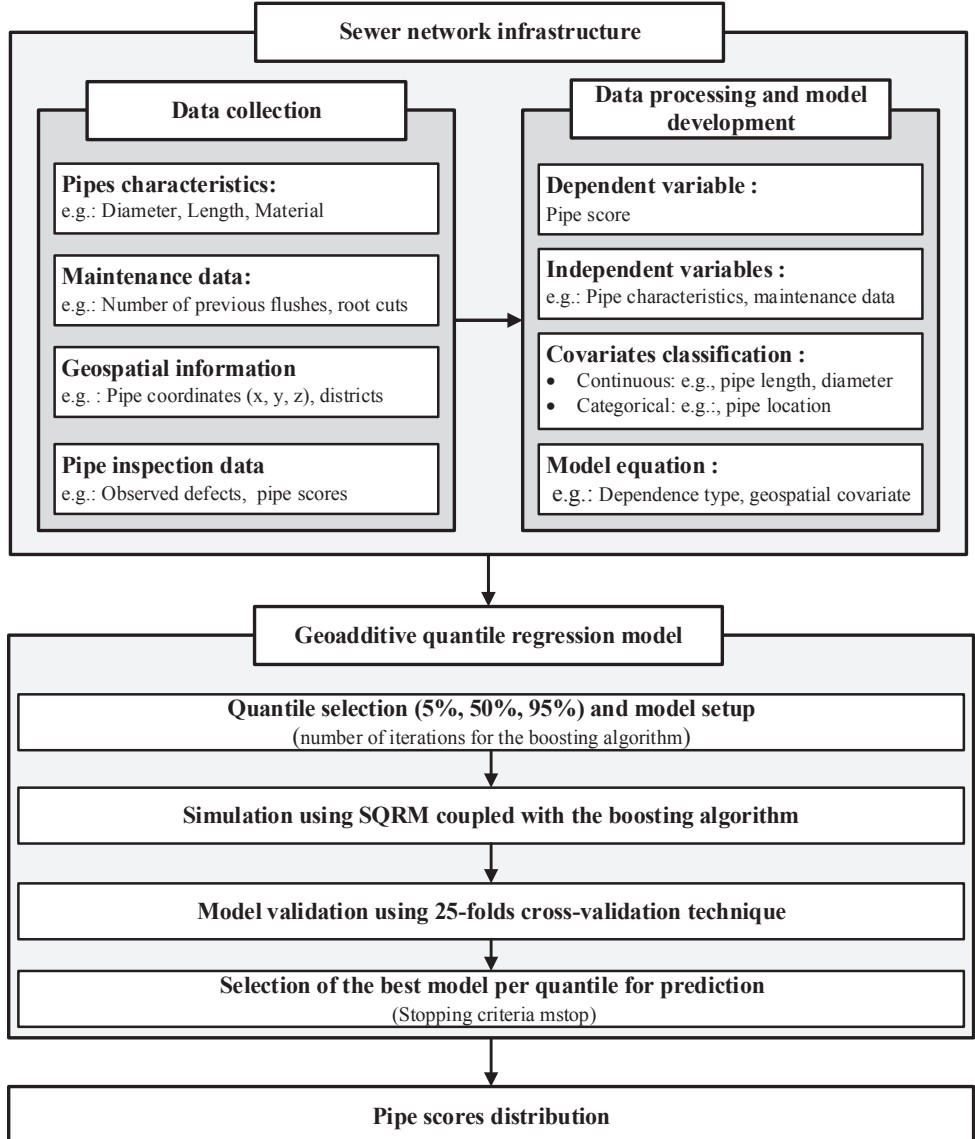

**Figure 1.** Proposed framework for geoadditive quantile regression model.

## 2.1. Geoadditive Quantile Regression

Flexible models better represent the complex nonlinear relationships between the regressors and the response variables in natural processes. These models allow the addition of noise that account for over dispersion caused by unobserved heterogeneities and possible autocorrelation. Assume the observations $(Y_i, x_i, z_i)$, $i = 1, \ldots, n$ are elements of the inspection database where $Y_i$ represents the pipe segment score (continuous), $x_i = (x_{i1}, \cdots, x_{iq})$ and $z_i = (z_{i1}, \cdots, z_{ip})$ are vectors of categorical (e.g., material) and continuous covariates (e.g., age), respectively. The relation between the observations $Y_i$ and the predictions $\eta_i$ is defined as:

$$Y_i = \eta_i + \epsilon_i, \quad \epsilon_i \sim F(\epsilon_i | \theta), i \in I \tag{1}$$

where $i$ is a subset of $\{1, \ldots, n\}$; that is, not necessary all latent $\eta_i$ are observed through the data $Y_i$. $F$ is an unknown distribution function of error terms $\epsilon_i$ and may depend on some additional parameter $\theta$.

Given a fixed and known quantile $\tau \in (0,100\%)$ and assuming the $\tau$th quantile of the error in Equation (1) is zero, i.e., $(F^{-1}(\tau|\theta) = 0)$, the general additive quantile regression model is expressed as [45].

$$\mathbf{Q}_{(Y_i|x_i,z_i)}(\tau|x_i,z_i) = \eta_{\tau_i} = \mathbf{x}_i^T\beta_{\tau_i} + \sum_{j=1}^{p} f_{\tau_j}(z_{ji}) + \mathbf{b}_{\tau_i} \tag{2}$$

where $\mathbf{Q}_{(Y_i|x_i,z_i)}(\tau|x_i,z_i)$ is a conditional quantile response, $\eta_{\tau_i}$ is the semi-parametric predictor, $\beta_{\tau_i}$ is the vector coefficient of the parametric component, $f$ is the vector of smoothing functions for the nonparametric component and $\mathbf{b}_{\tau_i}$ is an unstructured random effect accounting for overdispersion caused by unobserved heterogeneity or for correlation in longitudinal data [40].

In Bayesian framework, appropriate prior distributions (e.g., diffuse priors are considered for the linear effects (here applied to both discrete (categorical) and continuous covariates), first and second order random walk priors are used for continuous covariates and the structured spatial effect is modeled through GMRF specified as an intrinsic conditional autoregressive model) are defined and applied [9,42]. The parametric component in Equation (2) is estimated using the ordinary least squared algorithm. The nonlinear smooth functions $f_{\tau_j}$ (Equation (3)) are approximated through penalized polynomial splines (P-splines) of degree $l_j$ represented as a linear combination of $m_j = h_j + l_j - 1$ B-splines basis functions $\mathbf{B}_{j,k}$ evaluated at prespecified knots $z_{j,min} = \xi_{j1} < \xi_{j2} < \cdots < \xi_{j,h_j} = z_{j,max}$.

$$f_{\tau_j}(z_{ij}) = \sum_{k=1}^{m_j} \gamma_{\tau_{jk}}\mathbf{B}_{jk}z_{ij} \quad i = 1\cdots n; \quad j = 1\cdots p \tag{3}$$

where the coefficients $\gamma_{\tau_{jk}}$ are the amplitude that scale the B-spline basis functions $\mathbf{B}_{jk}$ to fit the observed data. Smoothness of the functions and accurate fit to observed data are obtained through the definition of high number of knots as well as the imposition of penalties to adjacent B-splines [9,46]. The interactions between the covariates are defined as bivariate functions as explained in [47]. The conditional mean obtained for $\tau = 50\%$ in Equation (2) is the Structured Additive Regression model [43,48]. Information about the geospatial location of the wastewater pipes is added to Equation (2) to build the geoadditive SQRM as:

$$\mathbf{Q}_{(Y_i|x_i,z_i)}(\tau|x_i,z_i) = \eta_{\tau_i} = \mathbf{x}_i^T\beta_{\tau_i} + \sum_{j=1}^{p} f_{\tau_j}(z_{ji}) + f_{geo,\tau}(S_i) \tag{4}$$

where $f_{geo}$ is spatial effect and has two components: the structured correlated effect $f_{struct}$ and the unstructured effect $f_{unstruct}$, specific to the place where the pipe is buried (local heterogeneities), i.e., $f_{geo} = f_{struct} + f_{unstruct}$. For each district $s \in 1, \cdots, S$ in the area of interest (e.g., in a city, different communities may represent districts), a separate regression coefficient is estimated to represent the correlated spatial effect $f_{struct} = (f_{struct}(S_1) \cdots f_{struct}(S_n))' = Z_{struct}\gamma_{struct}$. The design matrix $Z_{struct}$ is an incidence matrix whose entry in the $i$th row and $k$th column is equal to one if the observation i has been observed at location $k$ and zero otherwise. Spatially adjacent regions are assumed to share similar effects on the deterioration of pipes. This is achieved by defining the adjacency matrix $\mathbf{K}$ as follows:

$$\mathbf{K}[r,s] = \begin{cases} -1 & \text{if districts } s \text{ and } r \text{ are neighbors (i.e., } s \neq r,\ s \sim r) \\ |N(s)| & \text{if } s = r \\ 0 & \text{otherwise (i.e., } s \neq r,\ s \nsim r) \end{cases} \tag{5}$$

where $s \sim r$ denotes that the two regions $s$ and $r$ are neighbors and $N(s)$ is the total number of neighbors for region $s$. For example, in Figure 2, the selected district (Marlborough in the city of Calgary) has 8 neighbors

($N(s) = 8$). Further details about Markov Random Fields can be found in [49]. Information about the factors affecting the deterioration of sewer (e.g., sewerage type, load transfer, surface use, surface type, water main burst and leakage, ground disturbance, groundwater level, ground conditions, soil backfill type) are not systematically collected during inspection campaigns [9]. The addition of the geospatial location information allows to account for these factors [9,42]. The SQRM in Equation (4) is the final form of the equation used to represent the deterioration of sewer pipes in this research.

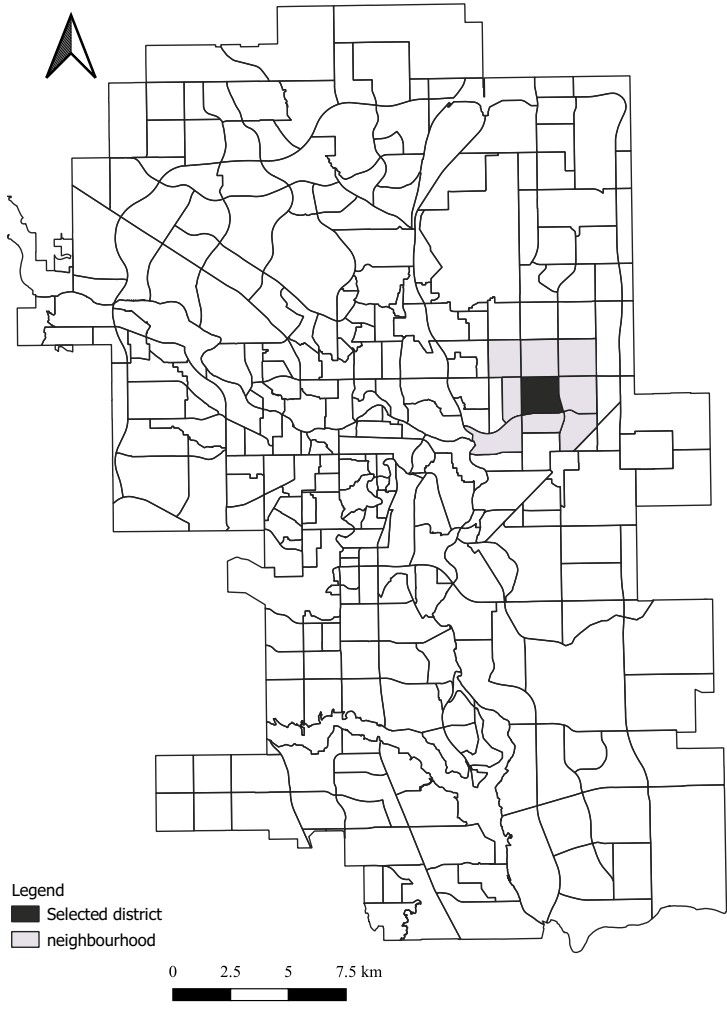

**Figure 2.** Example of neighborhood definition in Gaussian Markov Random field (GMRF).

For classic linear quantile, the regression coefficients are estimated by minimizing the asymmetrically weighted sum of absolute deviations [50]. The SQRM (Equation (4)) can alternatively be expressed as a minimization problem as [10]:

$$\underset{\eta_\tau}{\arg\min} \sum_{i=1}^{n} \rho_\tau (Y_i - \eta_{\tau i}) \tag{6}$$

where

$$\rho_\tau (y_i - \eta_{\tau i}) = \begin{cases} \tau |y_i - \eta_{\tau i}| & \text{if } (y_i - \eta_{\tau i}) \geq 0 \\ (\tau - 1)|y_i - \eta_{\tau i}| & \text{if } (y_i - \eta_{\tau i}) < 0 \end{cases}$$

is the "check function", i.e., a loss function for quantile regression problem that is appropriate for decision making perspectives [50]. The parametric component of the model is solved using the ordinary least square method.

## 2.2. Boosting Algorithm

The high number of covariates in the model (Equation (4)) led to the choice of boosting algorithm for variable and model selection [37,47]. The gradient descent (boosting) algorithm iteratively applies simple base learner procedures to residuals u obtained by differentiating the optimization criterion of interest (e.g., the loss function defined in Equation (6)) [47]. The stepwise increment of $\hat{f}_j^{[m]}$ (Algorithm 1) are small and the overall minimum is slowly approximated. At the same time, the additive structure of the resulting model fit is conserved since the final aggregation of the additive predictor and its single component are strictly additive. The algorithm selects the most informative base learners (model components) that minimize the empirical risk function. Since each base learner is composed of a single covariate, the variable selection is a very efficient process because the non-informative covariates will not be updated (i.e., they will never be the best fitting base learner in the optimization part of the algorithm) until the stopping criteria is reached. Thus, only influential base learners will appear in the final model. The step length factor $\nu$ is of minor effect on the accuracy of the estimators if a sufficiently small number is selected [10]. In the developed approach, K-cross-validation technique is applied. The data are split into K subsets where a single subset is kept for the validation and the other (K-1) subsets are used for the training. The goal of boosting algorithm in structured additive quantile regression is to find a solution to the expected loss optimization problem given by the following equation [51]:

$$\eta^* = \underset{\eta}{\operatorname{argmin}} \, \mathbb{E}[L(\mathbf{Y}, \eta)] \tag{7}$$

where $\mathbf{Y}$ is the response and $\eta$ is the predictor of a regression model. $L(\mathbf{Y}, \eta)$ corresponds to the loss function defined in Equation (6). From the observations $(\mathbf{Y}_i, \mathbf{x}_i, \mathbf{z}_i)$, with $i = 1, \cdots, n$ the number of observations; the expected loss in Equation (6) can be replaced by the following empirical risk:

$$\frac{1}{n}[L(\mathbf{Y_i}, \eta_i(\mathbf{x}_i, \mathbf{z}_i))] \tag{8}$$

where $\mathbf{x}_i, \mathbf{z}_i$ are the complete covariate vectors of observations.

Boosting algorithm advantage is able to combine the model fitting with the intrinsic variable selection and model choice through a functional gradient descent minimization framework [10,47,52,53]. In addition, in the presence of many predictors, the boosting algorithm shows high efficiency in the selection of variables compared to classical selection schemes [53]. The reduced number of parameters to be set by the user, including the number of iterations and the step-length factor (Algorithm 1), makes it a good tool to propose to municipalities for the modeling of the deterioration of wastewater pipes. Friedman [51] demonstrated that a small step-length is beneficial and never yields worse predictions using boosting algorithm. Thus, the step-length is set once. As explained in [51] and represented in Algorithm 1, the number of iterations will be changed after different runs until a better convergence is obtained. The optimal stopping criteria is determined using cross-validation and represents the optimal number of iterations needed to reach the minimum risk defined in Equation (8). The K-cross validation assists in avoiding the overfitting. The value of the stopping criteria (mstop) should be between 1 and the selected number of iterations.

---

**Algorithm 1:** Component-wise functional gradient descent boosting algorithm for structured additive quantile regression.

---

Initialization;

After selection the stopping criteria mstop, the definition and initialization of the predictor are the next steps to consider.;

From Equation (4), the type of interaction of each covariate is defined and the model built.;

Set the iterator m = 0 and the initialization of the response is done from suitable starting values such as the median of the response values $Y_1 \cdots Y_n$ as offset, i.e., $\eta_i^{[0]} = argmin_c \sum_{i=1}^{n} \rho_{0.5}(Y_i - c)$, and $\hat{f}_j^{[0]} = 0$ for $j = 1 \cdots p$;

Iteration;

**while** $m < mstop$ **do**

> Negative gradient;
>
> m = m + 1 (increment the iterator);
>
> Compute the negative gradient residuals of the loss function evaluated at the predictor values $\hat{\eta}_i^{[m-1]}$ (previous iteration estimated value);
>
> $$u_i^{[m]} = -\frac{\partial}{\partial \eta} L(Y_i, \eta) \mid_{\eta = \hat{\eta}_i^{[m-1]}} \quad i = 1 \cdots n$$
>
> For quantile regression boosting;
>
> $$u_i^{[m]} = -\rho_\tau' \left(Y_i - \hat{\eta}_i^{[m-1]}\right) = \begin{cases} \tau & Y_i - \hat{\eta}_i^{[m-1]} \geq 0 \\ \tau - 1 & Y_i - \hat{\eta}_i^{[m-1]} < 0 \end{cases}$$
>
> Estimation;
>
> Fit all the base learners separately to negative gradient residuals and obtain for each base learner, the $\hat{g}_j^{[m]}$ estimators of $\hat{f}_j^{[m]}$.;
>
> Run an optimization process to estimate the best fitting base learner $g_{j*}$ that minimizes the loss function below:
>
> $$j^* = argmin_j \left[ \left(u^{[m]} - \hat{g}_j^{[m]}\right)^T \left(u^{[m]} - \hat{g}_j^{[m]}\right) \right]$$
>
> where $u^{[m]} = (u_1^{[m]}, \cdots, u_n^{[m]})^T$ is the vector of gradient residuals in the current iteration
>
> Update;
>
> Compute the update for the nest fitting base learner
>
> $$\hat{f}_{j*}^{[m]} = \hat{f}_{j*}^{[m-1]} + v\hat{g}_{j*}^{[m]}$$
>
> where $v \in [0, 1]$ is a given step length. Keep all the other effects constant, i.e., set $\hat{f}_j^{[m]} = \hat{f}_j^{[m-1]}$ for $d \neq d^*$ Compute the $\hat{\eta}_i^{[m]}$ for $i = 1 \cdots n$

**end**

---

## 3. Case Study

The city of Calgary is situated in Southern Alberta, Canada and has a population of 1.2 million people. Its wastewater system is composed of 76,380 sewer pipes representing a total length of 5545.9 km (Figure 3). These pipes connect three wastewater treatment plants (Fish Creek, Pine Creek and Bonnybrook) to residential homes, commercial, industrial and public buildings. The database of the city indicates 5.24% (299.5 km) of pipes have their age above 65 years with 3.4% (188.6 km) above 100 years old. Different pipe material has been used to build the system: asbestos cement (AC), concrete (CON), brick (BR), vitrified clay (VCT), steel (ST), cast iron (CI), corrugated metal pipes (CMP); polyvinyl chloride (PVC), polyethylene (PE) sewer pipes. Before 1945, most of the pipes were clay or metallic. From 1945 to date, a predominance of plastic and cementitious pipes is observed in the data. In this study, pipe materials are grouped into: (1) Cementitious: CON, AC; (2) clay: BRICK, VCT; (3) metallic pipes: ST, CI, CMP; and (4) plastic: PE, PVC. The plastic and cementitious pipes represent 89.4% of the total pipes length in the system

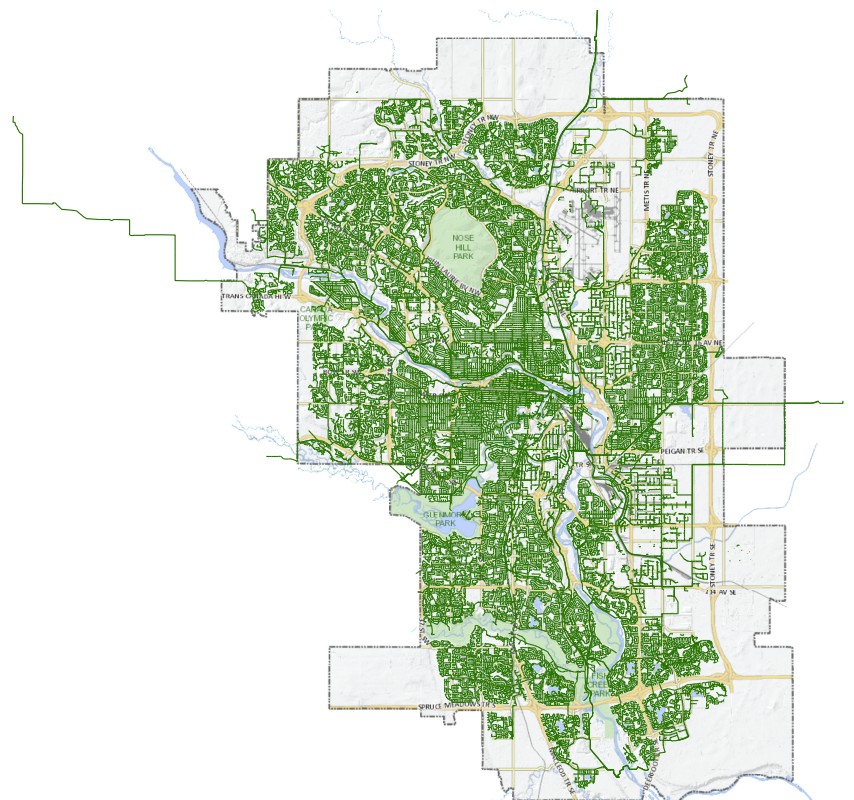

**Figure 3.** Calgary sewer pipe network.

### 3.1. Inspection Data Collection and Processing

Since early 1990, an inspection program using closed circuit television (CCTV) technique has been implemented in order to prioritize further inspection, maintenance and replacement of pipes. The selection of pipes for inspection is based on their age, material and the expert knowledge (e.g., out of 76,380 pipes in the city, 12,662 (1188.2 km i.e., 16.6%) have been inspected among which 68.2% are cementitious, 17.2% are clay, 10.1% are metallic and 4.5% are plastic pipes. Starting in 1940, the replacement of Ac and metallic pipes with concrete and plastic has been intensified in 1960. Figure 4 shows the age distribution of cementitious pipes that is varying between 0 and 55 and for plastic between 0 and 45 years. The installation

period has been reported as important parameter that affect the sewer condition. It integrates the age, the quality of the material used and the workmanship [54,55].

The observed defects data, for the city of Calgary in Canada, have been graded following the WRc protocol [44]. In this protocol, defects are categorized and assigned a score describing their severity and orientation. The pipe segment grade is given based on its highest defect score that is converted into a crisp number varying from 1 (best) to 5 (likely to fail) representing the Internal Condition Grading (ICG).

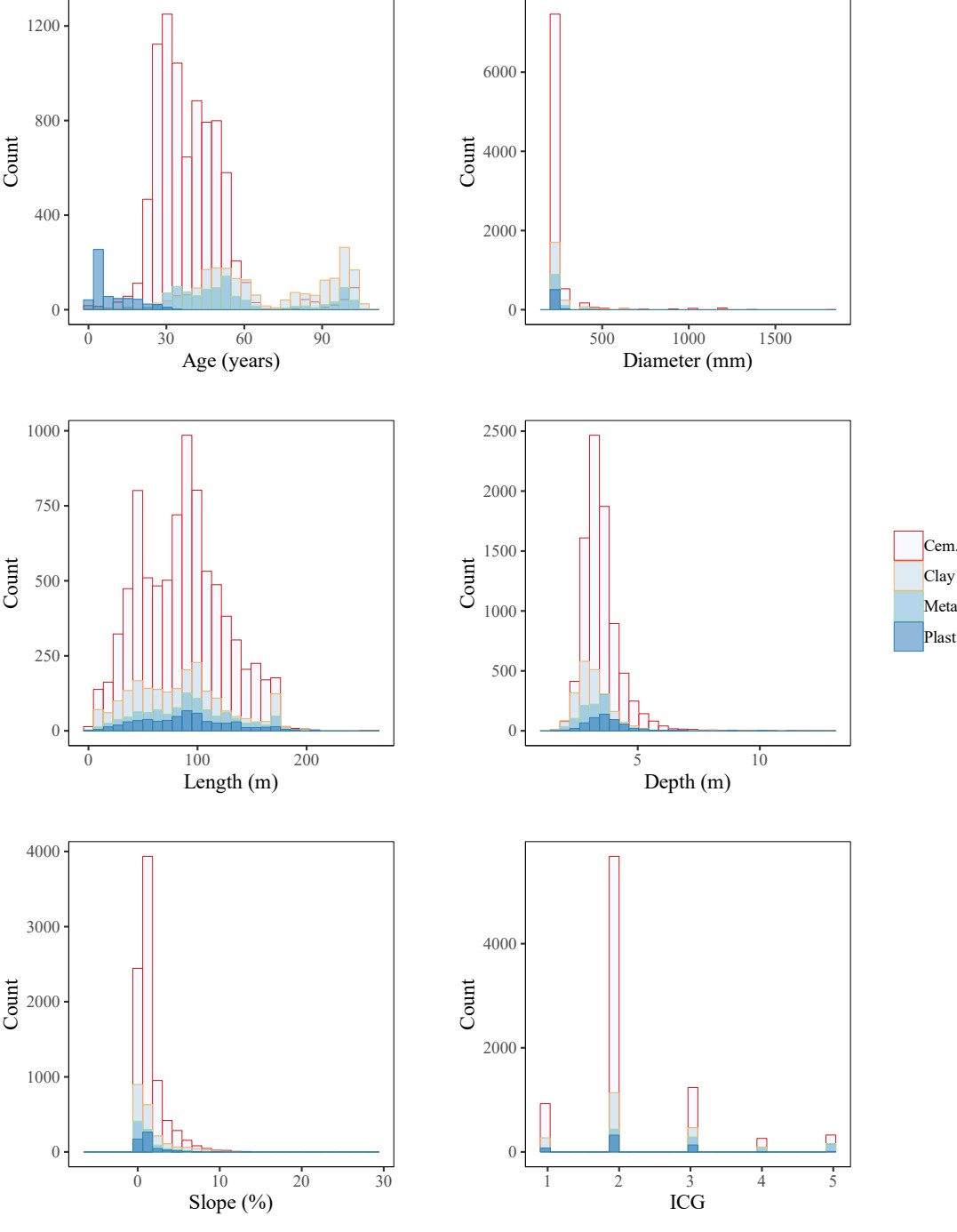

**Figure 4.** Paired characteristics of inspected pipes (City of Calgary).

*3.2. Factors Selection and Model Development*

Table 1 gives a brief description of the factors used in this study and assumed to influence the sewer pipe condition deterioration in the city of Calgary. These factors have been selected based on their availability from the database. Maintenance operations increase the pipes service life. However, some maintenance techniques (e.g., flushing, cleaning) may initiate erosion in the sewer material that will evolve with time. Maintenance factors are considered in the presented model.

**Table 1.** Covariates' description.

| Covariates (Physical) | Description |
| --- | --- |
| Material | Designed material of pipes (categorical: 1 = cementitious, 2 = clay, 3 = metallic, 4 = plastic) |
| Age | The difference between the inspection date and the installation date (continuous) |
| Length | The manhole to manhole distance inspected (continuous) |
| Diameter | size of the pipes (continuous) |
| Depth | mean distance from the crown of the pipe to the ground surface (continuous) |
| Slope | Angle between the pipes axis with the horizontal (continuous) |
| rservs | Number of residential connections to the pipe (continuous) |
| cservs | Number of commercial buildings connected to the pipe (continuous) |
| **Covariates (Maintenance)** | |
| Flushes | Number of flushes done previously on the sewer pipe (continuous) |
| repairs | Number of previous repairs on the sewer pipe (continuous) |
| Rcuts | Number of times, the roots have been cut from inside the pipe (continuous) |
| Backups | Number of times, there have been water backups in the sewers (continuous) |
| Degrease | Number of time, the maintenance team have degreased the sewer pipe (continuous) |
| Cleaning | Number of time, the sewer pipes have been cleaned (continuous) |
| **Covariates (Environmental)** | |
| Geospatial location | The geographic location of the sewer pipe. While it is widely accepted that pipes having the same characteristics deteriorates at the same pace, there is a variation in the deterioration across pipes in the same cohort but affected by different environmental covariates such as the nature of the soil they are buried in, the groundwater fluctuation, soil compaction, traffic on the ground, activities in the vicinity. These unobserved and unknown data have correlated and uncorrelated effects on the structural response |

Penalized splines with second order difference penalty was assumed as prior distribution for the continuous variables (e.g., [10,47]) and ordinary least squares approach is applied for the linear effects (e.g., [10,47,56]). Only the material covariate is considered as a categorical variable in the modeling and its fixed effect is assessed for each material type. The number of knots is set to 40 for the following covariates: diameter, age, spatial effect, flushes, cleaning, repairs, degrease and 20 for the others.

The geospatial information is captured through the location of pipes in districts. All the pipes of a district are assigned the same identification number, which means they are affected by the same unobserved covariates. The city of Calgary has 301 communities representing the districts (id) (Figure 2).

*3.3. Response Variable*

The response variable (score of pipes) is a continuous variable (following WRc protocol) that has been observed on the pipe segment during inspection. The geoadditive SQRM (Equation (4)) is represented in the following form:

$$
\begin{aligned}
\eta_\tau = {} & a_1 \times (Age) + f(Age) + a_2 \times (rservs) + f(rservs) + a_3 \times (cservs) + \\
& f(cservs) + a_4 \times (Length) + f(Length) + a_5 \times (Diameter) + f(Diameter) + \\
& a_6 \times (Material) + a_7 \times (Flushes) + f(Flushes) + a_8 \times (Cleaning) + \\
& f(Cleaning) + a_9 \times (Degrease) + f(Degrease) + a_{10} \times (Backups) + \\
& f(Backups) + a_{11} \times (Roots) + f(Roots) + a_{12} \times (repairs) + f(repairs) + \\
& a_{13} \times (Replaced) + f(Replaced) + a_{14} \times (Slope) + f(Slope) + \\
& a_{15} \times (Depth) + f(Depth) + map(id, bnd = clgr) + rd(id, bnd = clgr)
\end{aligned}
\tag{9}
$$

where $a_i$ is a fixed parameter coefficient estimated for linear effect, $f$ is smooth nonlinear function expressing nonlinearity in the relationships, *map* is the structured effect of the geospatial information, *rd* is the unstructured geospatial effect. The boosting algorithm iteratively builds each quantile model by selecting and updating the more informative covariates. Model parameters are selected as follow: initial number of iterations mstop = 30,000 and the step length = 0.25. The cross-validation of the models is done using 25-folds cross-validation technique. The optimal model used for prediction correspond to the one obtained at the optimal stopping criteria. The selected covariates at this point are the most influential parameters for the quantile under analysis. The effect of the interaction between factors is included in the model as a smooth function $f$.

## 4. Results and Discussion

*4.1. Model Selected Covariates*

Several reasons may explain the appearance of extreme values (e.g., human errors in the interpretation of the inspection data, not updating the database after replacement, maintenance and repairs, other reasons that need to be investigated on the field). In the presented results, human causes of errors are excluded, and the assigned sewer pipes condition grades are assumed to reflect the real actual condition of pipes.

Three quantiles regression models (5%, 50% and 95%). are developed and evaluated in this study. For each model, the score of pipes is predicted and the influential factors identified. The 5% and 95% quantile models are important in the schedule of further inspections and maintenance planning.

The optimal number of iterations obtained after the cross validation is 3675 for the 5% quantile, 18,656 for the 50% quantile and 20,000 for the 95% quantile. At this value (e.g., 3675 iterations as shown in Figure 5), the variables that have been incorporated in the deterioration model are the most influential, i.e., the most informative (Table 2). Both types of relationships are iteratively included in the model. Thus, the final optimal model may have linear, nonlinear and geospatial effects of the covariates on the output ICG response as presented in Table 2.

Interactions between the covariates have been considered in the developed model (Equation (9)). However, no interaction has been found informative by the applied boosting algorithm. Different covariates have been found to have mixed (linear and a nonlinear) effect to the pipe score. Table 2 shows the identified types of effects for all the included covariates in the model. Several studies, where the relationship between the pipe condition and the diameter is expressed in a linear form, found the diameter as an influential factor that affects the deterioration of sewer pipes [25,26,54,57–60]. However,

none of these studies gave details on how this covariate acts upon the sewer pipe scores. It is shown here that the diameter of the pipe influences linearly the observation of extreme high score values (95% quantile) of sewer pipes deterioration only (Figure 6a). The diameter does not influence the 5% quantile and 50% quantile. The length covariate is the only characteristic of sewer pipe segments that influences nonlinearly the 5% quantile (Figure 6b) and does not affect the 50% and 95% quantiles. It has been reported in previous studies that this covariate has an influence on the ICG. However, the dependence type and the affected quantile are not described [15,57].

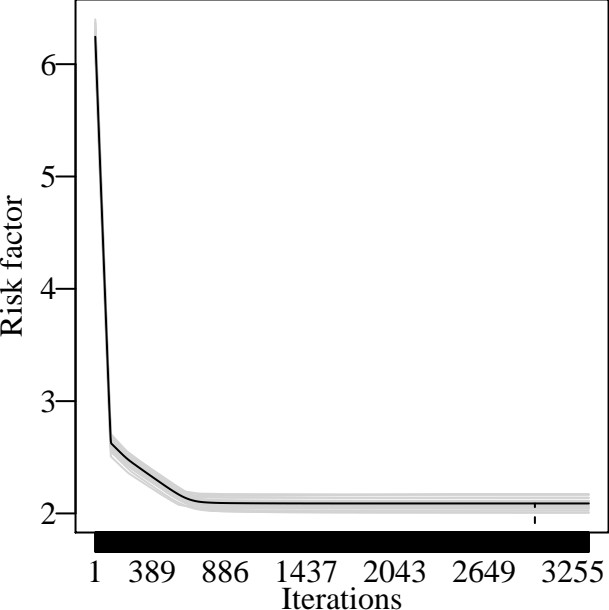

**Figure 5.** Optimal number of function evaluation to avoid overfitting in the 5% quantile.

**Table 2.** Influential covariates for each selected percentile.

| Covariates | 5th Percentile | | 50th Percentile | | 95th Percentile | |
|---|---|---|---|---|---|---|
| | Linear | Nonlinear | Linear | Nonlinear | Linear | Nonlinear |
| Age | - | - | x | x | x | x |
| rservs | - | x | x | x | | x |
| cservs | - | x | x | - | x | - |
| Length | - | x | - | - | - | - |
| Diameter | - | - | - | - | x | - |
| Material | x | - | x | - | x | - |
| Flushes | - | x | x | x | - | x |
| Cleaning | - | x | - | x | - | - |
| Degrease | - | x | - | x | - | - |
| Backups | - | x | - | x | - | - |
| Roots | - | x | - | x | - | - |
| repairs | x | x | x | x | x | x |
| Replaced | - | - | X | - | x | - |
| Slope | - | x | - | x | - | - |
| Depth | - | x | - | x | - | - |

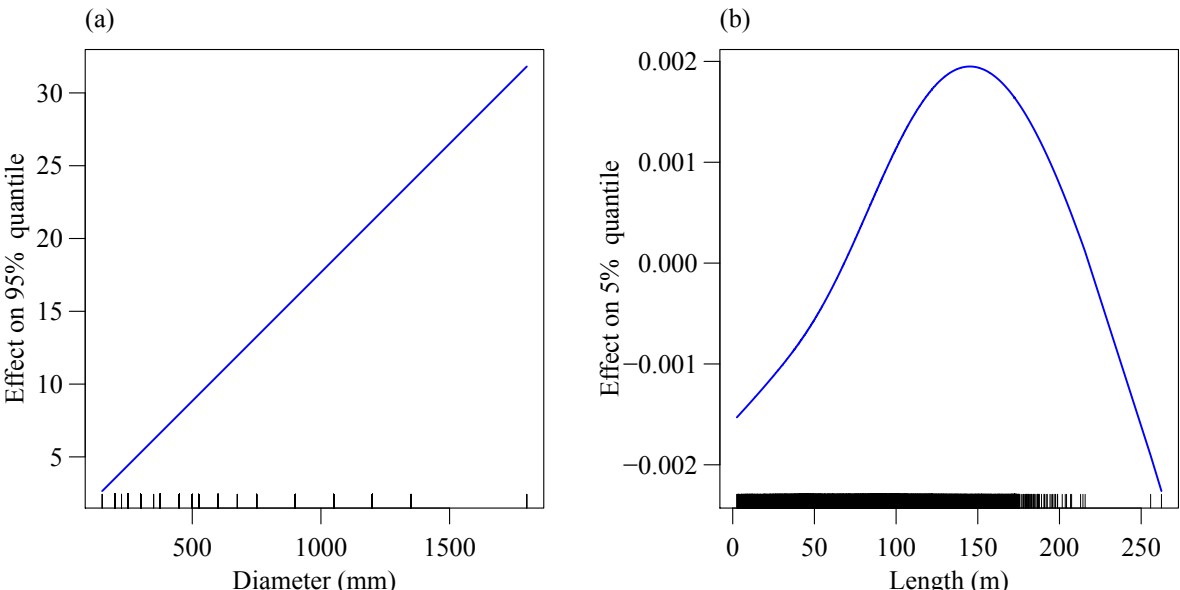

**Figure 6.** (**a**) Linear effect of the diameter on the 95% quantile; (**b**) nonlinear effects of the length covariate on the 5% quantile.

The age covariate only affects the 50% and 95% quantiles. Municipalities mostly use age covariate to support their decision making for inspections and budget planning. The following studies highlighted the influence of the age variable on the condition of sewer pipes [25,26,57,58,61,62]. When analyzing sewer pipes in the 5% quantile, the age covariate may be misleading as decision variable because it does not affect the 5% quantile. However, the 50% and 95% quantiles are both affected linearly and nonlinearly by the age covariate. Figure 7a,b shows both effect of the age covariates on the 95% quantile.

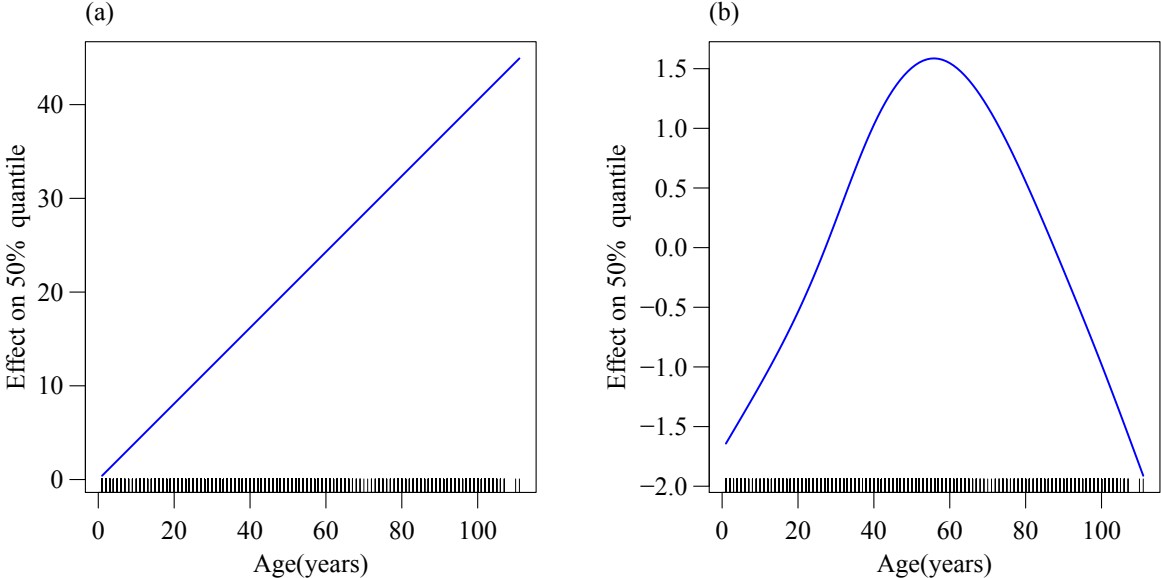

**Figure 7.** (**a**) Linear and (**b**) nonlinear effect of the age covariate on the 95% quantile.

The 5% quantile is affected by most of the independent variables except the following: Age, diameter and number of previous replacements. However, the 50% quantile (expected mean regression) is not affected by the diameter and length variables. Finally, only three maintenance covariates influence the 95% quantile predicted ICG response including the number of previous flushes flushes that have a nonlinear effect, the number of previous repairs with both types (linear and nonlinear) of relationships and the linear effect of the number of previous replacements.

The values of the covariate's effect in the 5% quantile are less than one. Both covariates have either negative influence (meaning the pipes are in the resistance period, i.e., a newly correctly installed pipe is in the flat part of the bath-tube shape deterioration pattern [63] or positive but below one demonstrating a low influence of the covariate to the deterioration process. In the 50% and 95% quantiles, however, the effect of some covariates goes above 250 (e.g., effect of number of previous repairs in the 95% quantile).

The response variable is an additive combination of simple estimators (constant and varying parameters) representing the partial effect of the selected covariates to the deterioration process. In the case, both types of effects are identified as influential for a covariate, the resulting effect is a mixture of the individual effects that are not easy to be captured with a single type of relationship between the predictor and the output ICG variable. An example is given in Figure 8a,b, where both relationships (linear and nonlinear) for the age and repairs covariates have been selected by the boosting algorithm as influential factors. The resulting mixed effect ("combined effect") shows that the age effect (Figure 8a) in the 50% quantile is mostly controlled by the linear part. However, the nonlinear part should be included to get a precise representation of the overall covariate effect. The repairs covariate effect Figure 8b is also a mixture of both linear and nonlinear effects when the number of previous repairs is below 4. However, after 4 repairs of a pipe, the repairs' effect is led by the nonlinear component.

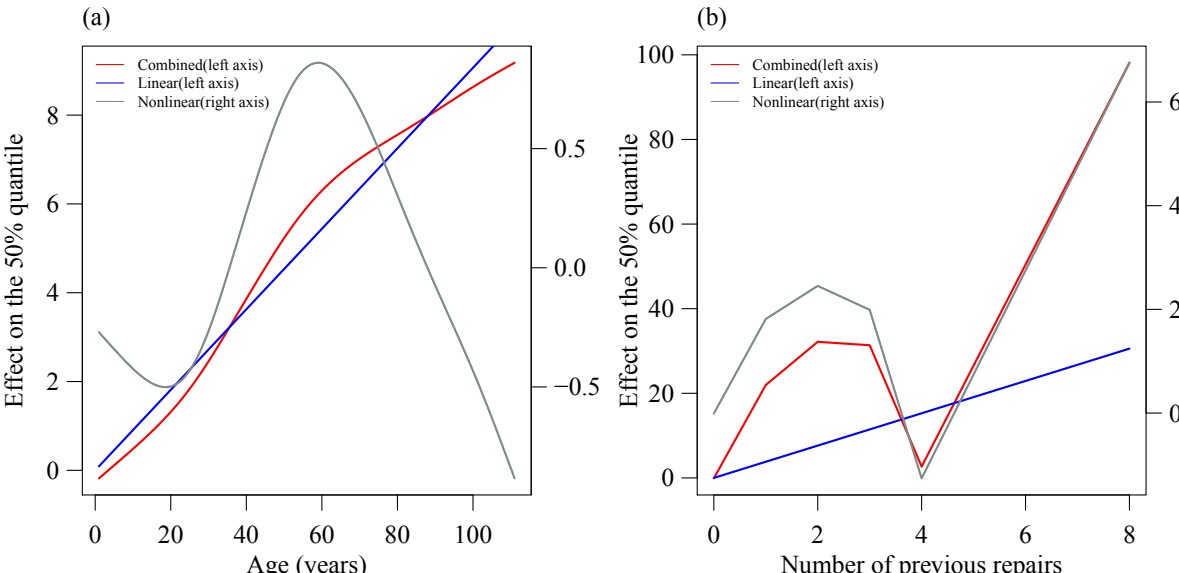

**Figure 8.** Combined effects of (**a**) age and (**b**) repairs covariates on the 50% quantile.

The inclusion in the model of either linear or nonlinear effect is in some cases the source of errors in the predictions of the output ICG.

### 4.2. Covariates' Effects on the Response Variable

Both quantile responses are influenced linearly and nonlinearly (Figures 6–8). The 5% quantile has the highest number of nonlinear factors' effects compared to the 50% and 95% quantiles. The type of dependence (linear and nonlinear) between the response variable and the predictors added to their effect's ranges describe the complexity of the wastewater deterioration process. Only the repairs covariate has both types of relationships (linear and nonlinear) with the pipes scores within all the selected quantile. Figure 8b shows both effects for the 50% quantile model. The 50% quantile, i.e., the normal (usual) expected mean regression model, gives a balance between the linear and the nonlinear relationships. Finally, the extremes observed high scores (95% quantile) are caused by only 8 factors (age, the number of residential connections, the number of commercial connections, diameter, material, the number of previous flushes, repairs and replacement) that influence linearly and/or nonlinearly the 95% quantile (Table 2).

Material, flushes and repairs covariates are the only factors that influence all the three selected quantile regression models (Table 2). The partial effects of the material (Figure 9) agree with the age of the pipes in different types of material as represented in Figure 4. Metallic and clay pipes are the oldest types of material in the network and show the highest partial effects compared to the newly installed types (cementitious and plastic pipes). In the 50% quantile, plastic pipes show a high partial effect compared to cementitious pipes while the revers is observed in the 95 % quantile. In additive models, each covariate is fitted individually to the response score (Figures 10 and 11). The model minimizes the differences between the output score and its predicted value from each single covariate (see Equation (6)). Once a good fit is obtained for each covariate, the global output is a weighted sum of the fit of the covariates.

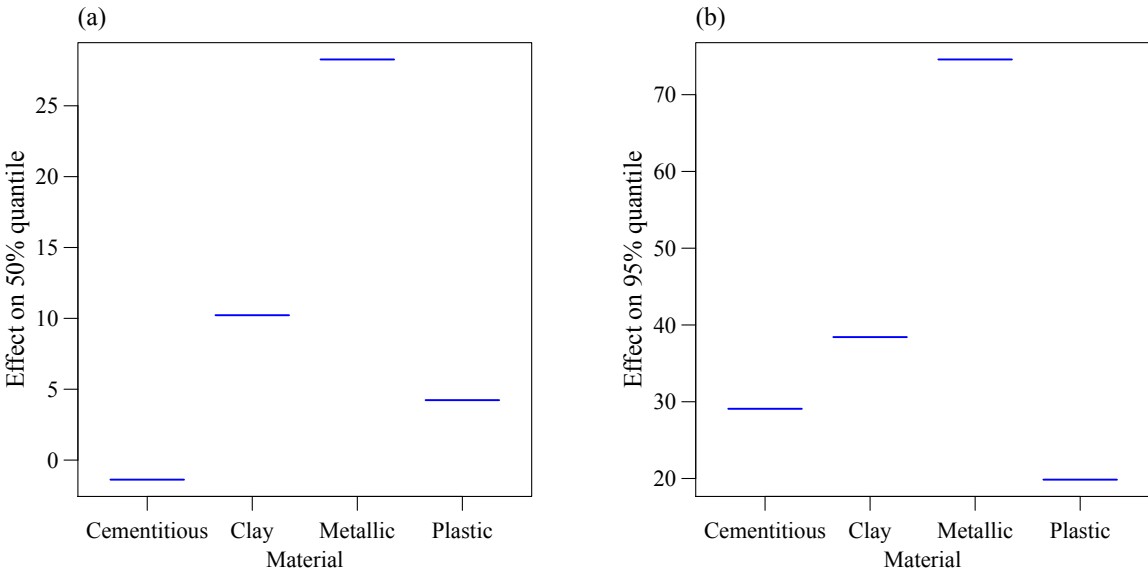

**Figure 9.** Material linear effect for (**a**) 50% quantile and (**b**) 95% quantile.

It can be seen from Figures 10 and 11 that the 5% quantile is not supported by much data. Thus, the decision maker should, in accordance with the map effect provided in the next section, investigate the delineated high effects areas to get more evidence (investigate new factors) that can support his decision-making process. However, for the 50% and 95% quantiles, Figures 10 and 11 show a good fit of the predictions over the observations. The developed quantile regression tool allow to capture a wide range of dependent variable.

The identified range is the band of uncertainty that the decision maker should consider when taking decisions about replacement, repairs or maintenance. The uncertainty range is skewed to the left with a long tail to the right (95% quantile). Thus, the decision maker can focus only on the 5% and 95% quantiles. In addition, the 95% quantile is the subset that has the highly risky pipes that should be prioritized for inspection and maintenance.

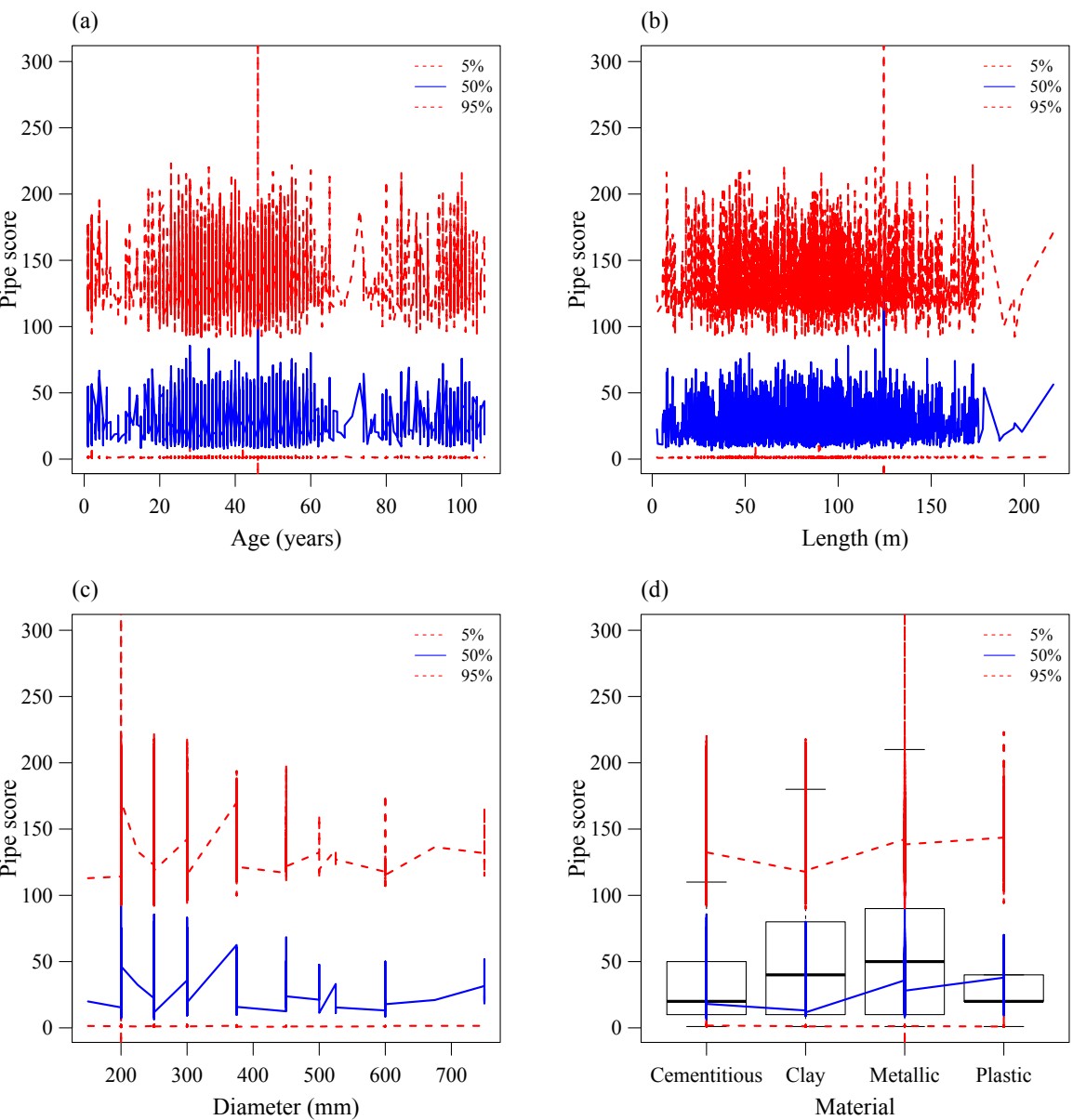

**Figure 10.** Physical characteristics covariates fitting to the response variables (**a**) age, (**b**) length, (**c**) diameter, (**d**) material.

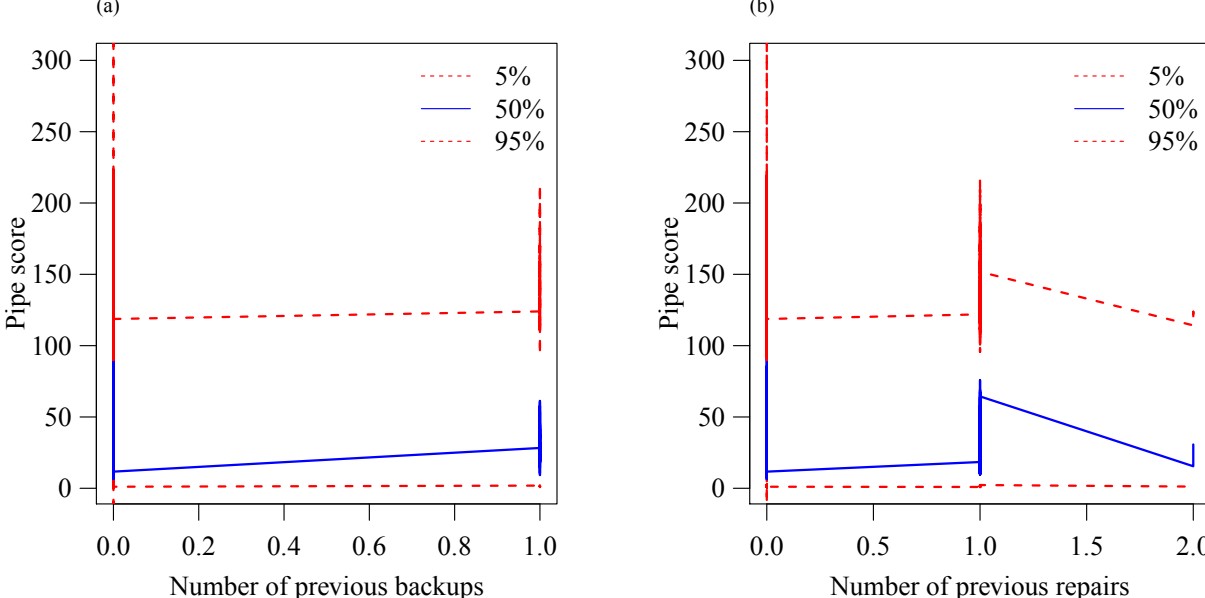

**Figure 11.** Example of maintenance covariates fitting to the output response (**a**) number of previous backups, (**b**) number of previous repairs.

### 4.3. Spatial Effects

Maps in Figure 12 are an important source of information to support early decision making for inspection needs and search of more explanatory variables that could better explain the undergoing deterioration of sewer pipes in each district of the city of Calgary. Furthermore, the geospatial effect represents the range of the rate of deterioration in each area of the city. The area around the airport shows the highest effect on the observed pipes condition scores (in the 50% quantile and 95% quantiles). Pipes in this area are cementitious. The causes of high deterioration rates in the airport area can be: (1) rigid pipes subject to intermittent surface loading due to frequent take-offs and landings of airplanes will have higher failure rate than plastic or other material types in the outer areas of the city, (2) the soil chemistry composition may have compound (e.g., $H_2S$) that reacts with the cementitious pipes and reduces their thickness. An in-depth investigation will likely reveal the main causes of the obtained high deterioration rate in the airport area. However, on the decision-making side, early decision making recommends an increase of inspection frequency in this area. The south west districts, where there are a mix of cementitious and plastic pipes, present also high deterioration rates for both 50% and 95% quantiles (Figure 12b,c).

The high deterioration rates in the north-east and south-west explain why city managers prioritized the replacement of metallic and clay pipes in the outer part of the city. The remaining metallic and clay pipes are buried in the center of the city of Calgary where the deterioration rate is not high. They are the oldest material type in the network (See Figure 4). The 50% quantile regression model (Figure 12b) highlights the same area of high influence (with a focus on the center of the network where more metallic and clay pipes are installed). The critical area (near the airport) is captured in the 95% quantile model. The analysis of quantiles allows to discover other causes (human activities) that may influence the deterioration pattern of sewer pipes in the system and delineate critical areas. Furthermore, the spatial effect maps show the existence of autocorrelation between the districts. The range values of the geospatial effects indicate that high geospatial effects are in the 95% quantile.

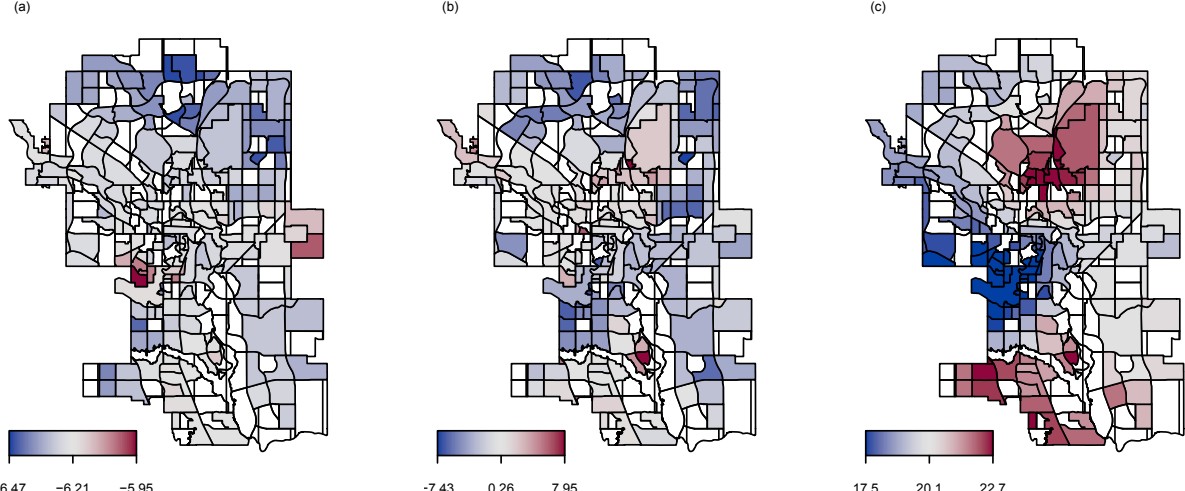

**Figure 12.** Spatial effects for (**a**) 5% quantile model, (**b**) 50% quantile and (**c**) 95% quantile.

### 4.4. Discussion

Quantile regression modeling is an effective way of addressing deterioration of sewer pipes. The proposed model is robust against outliers which are a frequent issue in the management of wastewater systems. It allows to elucidate factors that make extreme values to be observed. Moreover, it gives a credible interval in which the observed values are identified. This interval quantifies the band of uncertainty in the output variable for each independent variable (Figures 10 and 11). The band of uncertainty is key component of the decision-making process. Accuracy in the predictions is gained not only through acquisition of unknown information about contributing factors but also through a realistic representation of the complex dependence (linear, nonlinear) of the response (internal condition score) to the covariates. The obtained results agree with results obtained in previous studies [9]. For instance, the selected influential factors (see Table 2) have been found significant in many previous studies [62,64].

Kabir et al. [38] found in their application on the same case study that the age, the length and the rim elevation are the most significant factors for the cementitious pipes. For the clay pipes, their results show the importance of the age and the length. Finally, the plastic and metallic pipe deteriorations are influenced by the age and diameter. The analysis of the results in Table 1 show that the selected factors influence the deterioration of pipes but at different quantiles. For instance, the age is significant both (linearly and nonlinearly) for the 50% and 95% quantiles, not for the 5% quantile for both types of material. However, the diameter only influences the 95% quantile. The length is significant in the 5% quantile. The quantile approach highlights the effects of factors on different quantiles for both material types in the system. In addition, Balekelayi and Tesfamariam [42] applied nonlinear P-splines smooth functions to all the continuous factors and the only parametric factor was the material covariate. Their model found a root mean squared error (RMSE) equal to 17.33 while the RMSE for the 50% quantile in this study is equal to 17.46. In this study, the optimal 50% quantile model has a reduced number of covariates compared to the model developed in [9]. This study shows that it is efficient to consider only the most informative covariates in the quantile of interest. These covariates may differ from one quantile to another. Furthermore, the output maps highlight areas where further search of causing factors and pipes inspection should be prioritized. For inspection prioritization, the areas where no information have been recorded (district where only local heterogeneities are found) and districts with high geospatial effects are the priority areas for future inspection campaigns. The new information will be integrated in the developed model and the model will be updated. Regarding the quantiles, the 95% quantile is more informative

on pipes having a high deterioration rate and gives a better representation of critical zones than the 5% and 50% quantiles. From the analysis of the predicted ICG (Figures 10 and 11), it is observed that the distribution of the predicted response is skewed to the left and considering only the 50% and the 95% quantiles will be sufficient to give insights on the range of possible output responses.

## 5. Conclusions

In this paper, an individual deterioration model is developed using a Bayesian quantile inference with boosting algorithm for variable and model selection. The model considered the effects of physical, maintenance and environmental factors on the structural condition of sewer pipes using linear, nonlinear and geospatial effects on different quantiles (5%, 50% and 95%). The operator does not decide on the type of relationships between the model factors and the response. All possible combinations are incorporated in the model and an unsupervised selection of factors and model complexity is performed during the simulation by the model. Modeling quantiles allows the analysis and the identification of pipes in the tails of the condition distributions. Thus, different decisions should be made for pipes located in the same district and having the same characteristics. On the contrary, single decision (as suggested by several deterioration models) for both pipes (on the entire distribution) lead to premature replacement or the existence of high-risk pipes in the network. At a practical point of view, this research presents a powerful tool to orient and guide the prioritization of inspections in the network area. A map is a universal language tool, accessible to people from different personal and professional backgrounds. The geospatial effect maps on the deterioration of sewer pipes provide a sense of scale and distance of critical areas. The decision makers will have to decide on different priorities focusing on the 5% quantile and 95% quantiles for inspections. The rehabilitation plan may be based on the 50% quantile. Finally, this research showed that the pipe characteristic data representing the physical data (e.g., length, material, age, depth, slope), the maintenance data and the geospatial location of the pipes can efficiently represent a realistic deterioration model for all (inspected and uninspected) the pipes in the network. The proposed modeling approach allows the identification of factors that explain the difference occurring in the deterioration of pipes having the same characteristics and exposed to the same environmental factors. The defined uncertainty band is an important tool that should be efficiently used in maintenance planning and inspection prioritization. The presented methodology provides an accurate and effective tool that is based on reduced number of factors and through a proper representation of their effects to the response variable allows decision makers to analyze the entire distribution of the condition of pipes. Future research direction should include fine resolution in the geospatial location information. This information will allow to differentiate the effect of the environment on the pipes buried in the same district.

## 6. Future Research Needs

To support the proactive management of water utilities, there is a need of accurate deterioration models at reduced cost of data acquisition. The use of complex mathematical modeling to explain the deterioration process with available data is more than needed. Future research directions will use the proposed methodology in this study and reduced granularity for the geospatial location of the pipes (sub-district, roads). The objective is to have the geospatial location of pipes expressed in terms of its geographical coordinates.

**Author Contributions:** Conceptualization, methodology, writing—Original draft preparation and formal analysis, N.B.; review and editing, S.T. All authors have read and agreed to the published version of the manuscript.

**Funding:** This research received no external funding.

**Acknowledgments:** The authors acknowledge the financial support through the Natural Sciences an Engineering Research Council of Canada (RGPIN-2019-05584) under the Discovery Grant programs.

**Conflicts of Interest:** The authors declare no conflict of interest.

## Abbreviations

| | |
|---|---|
| SQRM | Structured Quantile Regression model |
| GMRF | Gaussian Markov Random Filed |
| ICG | Internal Condition Grading |
| EPA | Environmental Protection Agency |
| AC | Asbestos Cement |
| CON | Concrete |
| BR | Brick |
| VCT | Vitrified clay |
| ST | Steel |
| CI | Cast iron |
| CMP | Corrugated metal pipes |
| PVC | Polyvinyl chloride |
| PE | Polyethylene |

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
