# Peer review of "Geoadditive Quantile Regression Model for Sewer Pipes Deterioration Using Boosting Optimization Algorithm"

_sustainability, doi:10.3390/su12208733_

Round 1
Reviewer 1 Report
OVERALL COMMENTS: This is a good paper. The authors applied a semiparametric bayesian geoadditive quantile regression approach to estimate the deterioration of wastewater pipe from a set of covariates that are set to affect linearly and nonlinearly response variable. In general, a paper should be organized as follows: Introduction and Background; Objectives; Methodology, Results and Discussions; Conclusions; and Recommendations for Future Research. The authors should try to address all these above headings, including clearly specifying contributions to the body of knowledge at the end.
ABSTRACT: Should clearly have 1-2 lines each for introduction, objectives, methodology, results, conclusions, and recommendations for future research. Line 3 - there is an extra "the" before response variable.
*I think "heading 0" needs to be removed.
INTRODUCTION: Consider briefing and specifying this section with regards to the topic, and adding more literature as per the below suggestions:
Kaushal, V., & Najafi, M. (2020). Comparative Assessment of Environmental Impacts from Open-Cut Pipeline Replacement and Trenchless Cured-in-Place Pipe Renewal Method for Sanitary Sewers. Infrastructures, 5(6), 48.
Malek Mohammadi, M., Najafi, M., Kaushal, V., Serajiantehrani, R., Salehabadi, N., Ashoori, T. (2019). “Sewer Pipes Condition Prediction Models: A State-of-the-Art Review.” Infrastructures, Vol. 64, Issue 4, MDPI.
Kaushal, V., Najafi, M., Love, J., and Qasim, S. R. (2019). “Microbiologically Induced Deterioration and Protection of Concrete in Municipal Sewerage System: Technical Review.” Journal of Pipeline Systems Engineering and Practice, ASCE, Vol. 11, Issue 1.
Kaushal, V. and Najafi, M. (2020). “Comparison of Environmental and Social Costs of Trenchless Cured-in-Place Pipe Renewal Method with Open-cut Pipeline Replacement for Sanitary Sewers.” Journal of Pipeline Systems Engineering and Practice, ASCE, Vol. 11, Issue 4.
Malek Mohammadi, M., Najafi, M., Kermanshachi, S., Kaushal, V., Serajiantehrani, R. (2020). “Condition Prediction Modeling of Sewer Pipes: A State-of-the-Art Review.” Journal of Pipeline Systems Engineering and Practice, ASCE, Vol. 11, Issue 4.
Malek Mohammadi, M., Najafi, M., Serajiantehrani, R., Kaushal, V. (2020). “Predicting Condition of Sanitary Sewer Pipes with Random Forest.” Proc. ASCE Pipelines 2020, San Antonio, TX.
Kaushal, V., Najafi, M., Serajiantehrani, R., Malek Mohammadi, M. (2020). “A Framework for Evaluation of Social Costs of Open-cut Pipeline Replacement for Sanitary Sewers.” Proc. ASCE Pipelines 2020, San Antonio, TX.
MATHEMATICAL FORMULATION: What is the source of all the equations? Please consider specifying it.
CASE STUDY: Needs to be worked on more so far as the presentation and explanation is concerned.
RESULTS: There is a need to better organize and present the results to make it easy for the reader to comprehend the fndings. Also, the authors need to specify how their results are coherent with the previous results and if not, how much deviations are there?
*Discussions could be added as a different main heading before the conclusions.
CONCLUSIONS: The flow and grammar need to worked on and summarized to come to the main point of the paper.
Recommendations for future research and contributions to the body of knowledge are missing.
Any notation/acronyms/abbreviations used in the paper should be listed at the very end.
Author Response
General comment: This is a good paper. The authors applied a semiparametric bayesian geoadditive quantile regression approach to estimate the deterioration of wastewater pipe from a set of covariates that are set to affect linearly and nonlinearly response variable. In general, a paper should be organized as follows: Introduction and Background; Objectives; Methodology, Results and Discussions; Conclusions; and Recommendations for Future Research. The authors should try to address all these above headings, including clearly specifying contributions to the body of knowledge at the end.
Response: Thank you for the comment. Our approach aims at reducing the number of headings. Thus, the introduction includes the Introduction, the background, and the objective of the paper. That is why it appears to be long. The mathematical formulation is the proposed methodology. The conclusion has future recommendation.
C1. ABSTRACT: Should clearly have 1-2 lines each for introduction, objectives, methodology, results, conclusions, and recommendations for future research. Line 3 - there is an extra "the" before response variable. .
Response: Thank you for your comment. The abstract has been modified as follow:
“Proactive management of wastewater pipes requires the development of deterioration models that support maintenance and inspection prioritization. The complexity and the lack of understanding of the deterioration process make this task difficult. A semiparametric Bayesian geoadditive quantile regression approach is applied to estimate the deterioration of wastewater pipe from a set of covariates that are allowed to affect linearly and nonlinearly the response variable. Categorical covariates only affect linearly the response variable. In addition, geospatial information embedding the unknown and unobserved influential covariates is introduced as a surrogate covariate that capture global autocorrelations and local heterogeneities. Boosting optimization algorithm is formulated for variable selection and parameter estimation in the model. Three geoadditive quantile regression models (5%, 50% and 95%) are developed to evaluate the band of uncertainty in the prediction of the pipes scores. The proposed model is applied to the wastewater system of the city of Calgary. The results show that an optimal selection of covariates coupled with appropriate representation of the dependence between the covariates and the response increases the accuracy in the estimation of the uncertainty band of the response variable. The proposed modeling approach is useful for the prioritization of inspections and provide knowledge for future installations. In addition, decision makers will be informed of the probability of occurrence of extremes deterioration events when the identified causal factors, in the 5% and 95% quantiles, are observed on the field.”
C2. *I think "heading 0" needs to be removed..
Response: Acknowledged, The Introduction is the section 1 after the abstract
C3. INTRODUCTION: Consider briefing and specifying this section with regards to the topic, and adding more literature as per the below suggestions:
- Kaushal, V., & Najafi, M. (2020). Comparative Assessment of Environmental Impacts from Open-Cut Pipeline Replacement and Trenchless Cured-in-Place Pipe Renewal Method for Sanitary Sewers. Infrastructures, 5(6), 48.
- Malek Mohammadi, M., Najafi, M., Kaushal, V., Serajiantehrani, R., Salehabadi, N., Ashoori, T. (2019). “Sewer Pipes Condition Prediction Models: A State-of-the-Art Review.” Infrastructures, Vol. 64, Issue 4, MDPI.
- Kaushal, V., Najafi, M., Love, J., and Qasim, S. R. (2019). “Microbiologically Induced Deterioration and Protection of Concrete in Municipal Sewerage System: Technical Review.” Journal of Pipeline Systems Engineering and Practice, ASCE, Vol. 11, Issue 1.
- Kaushal, V. and Najafi, M. (2020). “Comparison of Environmental and Social Costs of Trenchless Cured-in-Place Pipe Renewal Method with Open-cut Pipeline Replacement for Sanitary Sewers.” Journal of Pipeline Systems Engineering and Practice, ASCE, Vol. 11, Issue 4.
- Malek Mohammadi, M., Najafi, M., Kermanshachi, S., Kaushal, V., Serajiantehrani, R. (2020). “Condition Prediction Modeling of Sewer Pipes: A State-of-the-Art Review.” Journal of Pipeline Systems Engineering and Practice, ASCE, Vol. 11, Issue 4.
- Malek Mohammadi, M., Najafi, M., Serajiantehrani, R., Kaushal, V. (2020). “Predicting Condition of Sanitary Sewer Pipes with Random Forest.” ASCE Pipelines 2020, San Antonio, TX.
- Kaushal, V., Najafi, M., Serajiantehrani, R., Malek Mohammadi, M. (2020). “A Framework for Evaluation of Social Costs of Open-cut Pipeline Replacement for Sanitary Sewers.” ASCE Pipelines 2020, San Antonio, TX.
Response: Thank you for the proposed literature. As said at the first comment, this section do not only have the introduction but it has the background and the objectives. The following citations are added to the text:
- Mohammadi et al. 2020, Predicting Condition of Sanitary Sewer Pipes with Gradient Boosting Tree, Pipelines 2020 : Condition Assessment, Construction, Rehabilitation, and Trenchless Technologies.
- Malek Mohammadi, M., Najafi, M., Kaushal, V., Serajiantehrani, R., Salehabadi, N., Ashoori, T. (2019). “Sewer Pipes Condition Prediction Models: A State-of-the-Art Review.” Infrastructures, Vol. 64, Issue 4, MDPI.
See line 86-88 :“Boosting algorithm has been successfully86applied for variable selection and model choice to estimate the time to failure of water pipes [41]87and the classification of sanitary sewers [39]”
Removed lines 68-71, 60-63, 78-80 in the old manuscript.
C4. MATHEMATICAL FORMULATION: What is the source of all the equations? Please consider specifying it.
Response: All the sources are cited either at the beginning of the equation or in the text explaining why the equation is used.
C5. CASE STUDY: Needs to be worked on more so far as the presentation and explanation is concerned.
Response: Acknowledged, the case study introduction is rewritten as follow:
“The city of Calgary is situated in Southern Alberta, Canada and has a population of 1.2 million people. Its wastewater system is composed of 76,380 sewer pipes representing a total length of 5545.9 km (Fig. 3). These pipes connect three wastewater treatment plants (Fish Creek, Pine Creek and Bonnybrook) to residential homes, commercial, industrial and public buildings. The database of the city indicates 5.24% (299.5 km) of pipes have their age above 65 years with 3.4% (188.6 km) above 100 years old. Different pipes material has been used to build the system: asbestos cement (AC), concrete (CON), brick (BR), vitrified clay (VCT), steel (ST), cast iron (CI), corrugated metal pipes (CMP); polyvinyl chloride (PVC), polyethylene (PE) sewer pipes. Before 1945, most of the pipes were Clay or Metallic. From 1945 to date, a predominance of Plastic and Cementitious pipes is observed in the data.
In this study, pipe materials are grouped into: (1) Cementitious: CON, AC; (2) Clay: BRICK, VCT; (3) Metallic pipes: ST, CI, CMP; and (4) Plastic: PE, PVC. The plastic and cementitious pipes represent 89.4% of the total pipes length in the system.”
C6. RESULTS: There is a need to better organize and present the results to make it easy for the reader to comprehend the fndings. Also, the authors need to specify how their results are coherent with the previous results and if not, how much deviations are there?
Response: Thank you…The lines 331-341 in the discussion section give the reason why the methodology is proposed. In addition, to the comparison with previous studies regarding the lines 350-354 shows a comparison between the proposed approach and previous study applied on the same city.
C7. *Discussions could be added as a different main heading before the conclusions.
Response: We have chosen to keep this as a subsection of the results and discussion section
C8. CONCLUSIONS: The flow and grammar need to worked on and summarized to come to the main point of the paper.
Response: Thank you. The conclusion has been rewritten as follow: “In this paper, an individual deterioration model is developed using a Bayesian quantile inference with boosting algorithm for variable and model selection. The model considered the effects of physical, maintenance and environmental factors on the structural condition of sewer pipes using linear, nonlinear and geospatial effects on different quantiles (5%, 50% and 95%).
The operator does not decide on the type of relationships between the model factors and the response. All possible combinations are incorporated in the model and an unsupervised selection of factors and model complexity is performed during the simulation by the model.
Modelling quantiles allows the analysis and the identification of pipes in the tails of the condition distributions. Thus, different decisions should be made for pipes located in the same district and having the same characteristics. On the contrary, single decision (as suggested by several deterioration models) for both pipes (on the entire distribution) lead to premature replacement or the existence of high-risk pipes in the network.
At practical point of view, this research presents a powerful tool to orient and guide the prioritization of inspections in the network area. A map is a universal language tool, accessible to people from different personal and professional backgrounds. The geospatial effect maps on the deterioration of sewer pipes provide a sense of scale and distance of critical areas. The decision makers will have to decide on different priorities focusing on the 5% quantile and 95% quantiles for inspections. The rehabilitation plan may be based on the 50% quantile.
Finally, this research showed that the pipe characteristic data representing the physical data (e.g., length, material, age, depth, slope), the maintenance data and the geospatial location of the pipes can efficiently represent a realistic deterioration model for all (inspected and uninspected) the pipes in the network. The proposed modelling approach allows the identification of factors that explain the difference occurring in the deterioration of pipes having the same characteristics and exposed to the same environmental factors. The defined uncertainty band is an important tool that should be efficiently used in maintenance planning and inspection prioritization.
The presented methodology provides an accurate and effective tool that is based on reduced number of factors and through a proper representation of their effects to the response variable allows decision makers to analyze the entire distribution of the condition of pipes. Future research direction should include fine resolution in the geospatial location information. This information will allow to differentiate the effect of the environment on the pipes buried in the same district.”
C9. Recommendations for future research and contributions to the body of knowledge are missing.
Response: Acknowledged, Future direction needs section is added to the manuscript after the conclusion
C10. Any notation/acronyms/abbreviations used in the paper should be listed at the very end.
Response: Thank you. A list of acronyms is added at the end

Reviewer 2 Report
The authors write about the difficulties in managing the network because there is no complete information. And they use thesame data for modeling; whether the data was homogeneous? The model considers only the effects of physical, maintenance. There's no information of environmental factors.
No information on the environment in which the proposed solution was implemented.There is no example implementation allowing for verification of the results.
Author Response
General comment: The authors write about the difficulties in managing the network because there is no complete information. And they use the same data for modeling; whether the data was homogeneous? The model considers only the effects of physical, maintenance. There's no information of environmental factors. No information on the environment in which the proposed solution was implemented. There is no example implementation allowing for verification of the results.
Response: Thank you for your comment. The data gives the required information. However, the techniques of extracting the information and gaining in the knowledge differ. In this paper, we propose to use the geospatial location of the pipes as a covariate that embed the environmental factors that affect the deterioration of the pipes. This is expressed in Table 1 and the line 305 is rewritten as follow:
“The geospatial information (embedding the environmental factors) is captured through the location of pipes in districts. All the pipes of a district are assigned the same identification number, which means they are affected by the same unobserved covariates.”
The section Results and Discussion present the application of the proposed methodology to the wastewater pipe of the city of Calgary and the comparison with previous results that were based on the mean regression is discussed.

Reviewer 3 Report
The authors present an interesting piece of work that could be used in practical scenarios. However, there are some issues that should be clarified before publication.
-The authors should discuss the impact of erroneous input data on the robustness of the proposal, measuring how different errors can influence the final results.
-The study is devoted to a concrete city (Calgary). Do you have any agreement/chance with Calgary to use your proposal in real life?
-The conclusions of the paper could be somehow Calgary-biased. As you are only considering input data from a single city, your results can be overfitted to the particular case. Your conclusions would be much more solid if you could consider alternative case studies.
Minor issues:
-Please, remove section 0.
Author Response
General comment: The authors present an interesting piece of work that could be used in practical scenarios.
C1. The authors should discuss the impact of erroneous input data on the robustness of the proposal, measuring how different errors can influence the final results.
Response: Thank you for your comment. The data has an influence on the results and that is showed in the study. However, due to its objective, this study considered that the data as provided by the municipality is accurate and should not contain errors.
C2. The study is devoted to a concrete city (Calgary). Do you have any agreement/chance with Calgary to use your proposal in real life?
Response: The proposed technique can be applied to any other city where the required data are available. No, there is no agreement with the city of Calgary
C3. The conclusions of the paper could be somehow Calgary-biased. As you are only considering input data from a single city, your results can be overfitted to the particular case. Your conclusions would be much more solid if you could consider alternative case studies.
Response: Thank you for your comment. The conclusion is not biased. The conclusion shows how the analysis of the quantile in the deterioration process of any city can highlight pipes in the tails that should not be treated the same way as the pipes in the majority of the pipes having the same characteristics. In addition, the environmental covariate is adding more information that makes the deterioration model more accurate.
C4. Minor issues: -Please, remove section
Response: Thanks, removed.

Round 2
Reviewer 3 Report
The current version of the paper can be accepted for publication.